

# Systematic review assessing the effects of amendments on acidic soils pH in tea plantations

Zhenyu Yang[1],[*], Bo Yao[2],[*], Ronghui Li[3], Wenyan Yang[1], Dubin Dong[4],[5], Zhengqian Ye[2], Yuchun Wang[1] and Jiawei Ma[1],[2]

[1] College of Tea Science and Tea Culture, Zhejiang A&F University, Hang Zhou, China
[2] Key Laboratory of Soil Contamination Bioremediation of Zhejiang Province, State Key Laboratory of Subtropical Silviculture, Zhejiang A&F University, Hang Zhou, China
[3] Agricultural and Rural Bureau of Quzhou, Quzhou, China
[4] Zhejiang A&F University, Hang Zhou, China
[5] Central South University of Forestry and Technology, Changsha, China
[*] These authors contributed equally to this work.

## ABSTRACT

Soil acidification has emerged as a critical limiting factor for the sustainable development of the tea industry. In this article, a comprehensive review of 63 original research articles focusing on the impact of amendments on the pH in tea plantations soil was conducted. Through meta-analysis, the effect of applying soil amendments to increase the pH of tea plantation soil and its influencing factors were investigated. The results revealed that lime had a significant impact, increasing the pH by 18% in tea plantation soil, while rapeseed cake had a minimal (2%) effect. It was observed that as the quantity of amendments and pH levels increased, so did their impact on the pH of tea plantation soil. Subgroup analysis within biochar showed varying effects, depending on soil pH, with an 11% increase in acidic soil. Among these amendments, biochar produced at pyrolysis temperature ranging from 501–600 °C and derived from animal waste demonstrated significant effect on increasing soil pH in tea plantations by 9% and 12%, respectively. This study offers valuable insights into improving and ensuring the health and sustainability of tea plantations.

## INTRODUCTION

China is an important tea-producing country, boasting the largest tea plantation area in the world. Tea plant (*Camellia sinensis*) has become a vital cash crop and primary export commodity in China (*Yan et al., 2018*; *Zhang et al., 2023*). Tea plants prefer acidic soil conditions, with an optimal soil pH range for tea plant growth is 4.5–5.5 (*Ruan et al., 2007*). Nevertheless, approximately 46% of tea plantation soil in China has a pH below 4.5 (*Yan et al., 2020*), resulting in an escalating issue of soil acidification within Chinese tea plantations which is progressively intensifying (*Zeng et al., 2017*). The primary contributors to such acidification include improper fertilizer application and the decomposition of organic acids released by the roots of the tea plant themselves (*Han et al., 2022*; *Yan et al., 2020*). This decrease in soil pH will inevitably impact soil characteristics by disrupting

Corresponding authors
Yuchun Wang,
ycwang0201@126.com
Jiawei Ma, jiawma@zafu.edu.cn

chemical processes, increasing levels of aluminum (Al) and manganese (Mn) toxicity, and leading to leaching loss of potassium (K), sodium (Na), calcium (Ca), magnesium (Mg), and other nutrients, ultimately diminishing overall soil fertility (*Alekseeva et al., 2011*; *Le et al., 2022*; *Fenn et al., 2006*). Furthermore, the population of beneficial microorganisms in the tea rhizosphere soil decreases (*Li et al., 2017*). Consequently, the tea tree's capacity to uptake soil nutrients diminishes, impacting its growth and lowering tea quality (*Goswami et al., 2017*; *Yang et al., 2022*). The acidic soil environment hampers the soil's capacity to adsorb heavy metal ions, activates them, enhances their solubility, mobility, and effectiveness, facilitates heavy metal uptake by tea trees, resulting in elevated heavy metal content in tea (*Bolan, Adriano & Curtin, 2003*). *Wen et al. (2018)* investigated the accumulation and transfer of heavy metals in the soil-tea system and revealed that as the soil pH value decreases, there is an increase in availability of heavy metals in tea trees. The issue of soil acidification in tea plantations poses a challenge to the sustainable development of tea plantations in China. Therefore, optimizing soil pH and controlling soil acidification are crucial for the sustainable development of tea plantations.

In recent years, numerous studies have investigated the impact of soil amendments on soil pH in tea plantations, such as biochar, lime, crop straw, rapeseed cake, animal waste. Biochar is a carbon-rich, porous, alkaline soil organic amendment derived from biomass through anaerobic pyrolysis (*Cheng et al., 2019*; *Li et al., 2023*). Many studies indicated that incorporating biochar into soils could improve soil structure, increase porosity, decrease bulk density, and enhance aggregation and water retention (*Baiamonte et al., 2015*). Furthermore, biochar has been found to increase soil electrical conductivity by 124.6% (*Oguntunde et al., 2004*), raise cation exchange capacity by 20% (*Laird et al., 2010*), and reduce soil acidity by 31.9% (*Oguntunde et al., 2004*). It has been widely used in agricultural soils to improve soil fertility and alleviate soil acidification (*Yang et al., 2021*). *Guo et al. (2023)* found that the application of biochar led to an increase in soil pH value, rising from 4.62 to 4.78 in a pot experiment. In addition to biochar, lime is widely used to ameliorate soil acidity (*Holland et al., 2018*). *Liu et al. (2023b)* observed in a field experiment that lime application in tea plantations soil resulted in a significant rise in soil pH from 4.22 to 4.70. Moreover, *Wang et al. (2013a)* reported from experiments that when various crop straw application rates led to increases in soil pH levels after incubation for 65 days: wheat straw by 0.37, rice straw by 0.22, peanut straw by 0.17, punk cake by 0.01, respectively, compared to the control. *Wang et al. (2018)* observed a significant increase in soil pH with the addition of quicklime and biochar, and this ameliorative impact was found to be more pronounced with higher doses of these amendments. Furthermore, *Wang et al. (2013b)* demonstrated in laboratory experiments that the application of rapeseed cake could also raise soil pH, and this increase was positively correlated with the quantity of rapeseed cake applied.

It is important to note that the influence of amendments on soil pH in tea plantations is not solely determined by their properties of these materials, but is also influenced by factors such as soil fertility conditions, climate conditions, variety characteristics and other environmental influences. While different researchers may reach divergent conclusions,
there remains a dearth of comprehensive quantitative assessment regarding the impact of applying amendments on soil pH in tea plantations. Thus, this study has collected and compiled published literature on the impact of amendments on the pH of tea plantations soil. It has analyzed the change in soil pH following the application of different amendments, and employed a meta-analysis method to further explore their effects. We are committed to establishing a technical guidance strategy of general significance for the enhancement of soil quality in tea plantations, and further providing scientific and technological support as well as a theoretical basis for the sustainable and healthy cultivation of tea plantations.

## MATERIALS AND METHODS

### Literature collection

We utilized the Web of Science (https://www.webofscience.com/wos), springer link (https://link.springer.com/), Google Scholar (https://scholar.scholar-xm.top/) and the China National Knowledge Infrastructure (https://www.cnki.net/) to investigate publications documenting the impacts of employing soil modifiers on the pH of tea plantations soil both domestically and internationally, up until June 2023. Specific search terms were used for literature screening, including "improvement materials", "biochar" or "crop straw" or "animal manure" or "lime" or "rapeseed cake fertilizer" and "tea plantations soil pH". Additionally, the reference lists of the retrieved articles were reviewed manually to identify additional relevant literature. In accordance with the data integration demands of the meta-analysis methodology and the objectives of this study, the retrieved original research articles was scrutinized using the following criteria: (1) The experimental conditions (*e.g.*, time, location, management measures) must be are clearly outlined; (2) the soil utilized in the experiment must specifically be from tea plantations; (3) the experiment should encompass at least one treatment set involving the application of soil amendments and a control group with non-amendments, while ensuring consistency in other experimental conditions; (4) at least three replicates of each treatment in the experiment; (5) data regarding the mean value and standard deviation of soil pH must be accessible throughout the experiment. Based on these screening criteria, a total of 63 original research articles and 366 sets of valid data were collected.

### Establishment of the database

The pH values were recorded for each article, including the mean for treatment and control, as well as the standard deviation (SD) or standard error (SE), and sample capacity (n). Standard deviation (SD) is an essential input variable in meta-analysis. Therefore, SD values were calculated in cases where the standard error (SE) for soil pH was presented in the manuscripts, using the equation for SD as provided by *Brett et al. (2017)*.

$$SD = SE^* \sqrt{n} \tag{1}$$

Data extraction from selected valid literature involved capturing information such as the title of the articles, the first author, the experiment location, experiment time, soil pH, soil nutrients, the type of soil amendments, the pH and application amount of the soil amendments. A database of the relationship between the modifiers and soil pH was
established through an Excel table. In building the database, data displayed in text and table form is directly extracted, while data displayed in graph form is using by GetData Graph Digitizer software. The unit of modifiers application is unified as t/ha. In pot or incubation experiments, unit conversion is 150,000 kg per 667 m$^2$ of tillage soil (*Zhang et al., 2016*). Soil pH values extracted by CaCl$_2$ and KCl were converted to deionized water extractable pH values by the following formula (*Nguyen et al., 2017*):

$$pH(H_2O) = 1.65 + 0.86pH(CaCl_2) \tag{2}$$

$$pH(H_2O) = 1.96 + 0.74pH(KCl) \tag{3}$$

## Data classification

According to "China Soil", the soil pH is classified as follows: extremely acidic soil (pH ≤ 4.5); strongly acidic soil (4.5 < pH ≤ 5.5); acidic soil (5.5 < pH ≤ 6.5); neutral soil (6.5 < pH ≤ 7.5); alkaline soil (pH > 7.5). The application rates for different amendments are categorized as follows: for biochar, the divisions are M ≤ 10 t/ha; 10 < M ≤ 40 t/ha; 40 < M ≤ 80 t/ha; M > 80 t/ha. For lime, the divisions are M ≤ 1.5 t/ha; 1.5 < M ≤ 4.5 t/ha; M > 4.5 t/ha. In the case of crop straw, the divisions are M ≤ 20 t/ha; 20 < M ≤ 40 t/ha; M > 80 t/ha. Finally, for animal waste, the divisions are M ≤ 4 t/ha; 4 < M ≤ 10 t/ha; M > 10 t/ha. The pH of the amendments was divided into three groups: pH <= 8, pH <= 10, and pH > 10. Then the biochar in the amendments continues to be classified in detail, and the types of biochar are roughly divided into four categories according to the raw materials of biochar: crop straw (*e.g.*, wheat straw, corn straw, rice straw), wood (*e.g.*, tea branches, bamboo), shell residue (*e.g.*, peanut shell, bamboo coconut shell) and animal waste (*e.g.*, pig manure, sheep manure). Regarding the pyrolysis temperature of biochar, if the literature gives a temperature range, it is classified as its average value. The pyrolysis temperature is divided into four ranges: low temperature (≤400 °C), medium temperature (401– 500 °C), medium high temperature (501– 600 °C), and high temperature (>600 °C).

## Statistical analysis

Meta-analysis is a statistical method that quantitatively synthesizes multiple research results. It has the capability to analyze comprehensively and quantitatively the effects of different influencing factors, exploring the relationship between results and clarifying the relative contribution of each influencing factor.

In this study, meta-analysis was utilized to compare the effect of soil amendments on soil pH, requiring the inclusion of quantitative experimental data for effect value indicators. The response ratio (RR) was calculated based on the natural logarithm of soil pH, with lnRR being determined using formula (*Hedges, Gurevitch & Curtis, 1999*) (4):

$$\ln RR = \ln (I/CK) = \ln I - \ln CK \tag{4}$$

In the formula, M represents the average soil pH of the treatment group after the application of modifiers, while CK represents the average soil pH of the control group without modifiers. Additionally, the coefficient of variation (V), weighted factor (W),

weighted response ratio (RR), the standard error (S) of RR, and its 95% confidence interval (CI) can be calculated as follows (*Luo, Hui & Zhang, 2006*):

$$V_{\ln RR} = \frac{SD_M^2}{n_M M^2} + \frac{SD_{CK}^2}{n_{CK} CK^2} \tag{5}$$

$$W_{ij} = \frac{1}{V} \tag{6}$$

$$RR_{++} = \sum_{i=1}^{m} \sum_{j=1}^{k_i} W_{ij} RR_{ij} \Big/ \sum_{i=1}^{m} \sum_{j=1}^{k_i} W_{ij} \tag{7}$$

$$s(RR_{++}) = \sqrt{\frac{1}{\sum_{i=1}^{m} \sum_{j=1}^{k_i} W_{ij}}} \tag{8}$$

In the formula, $SD_M^2$ and $SD_{CK}^2$ represent the standard deviation of the treatment group (with modifiers) and the control group (without modifiers), respectively. Similarly, $n_M$ and $n_{CK}$ denote the number of samples in the treatment group and the control group, respectively. The smaller the standard deviation of the effect value, the larger the weight assigned, and the weight response ratio (the percentage increase or decrease of the treatment relative to the control) and its 95% CI can be converted by $\exp(RR_{++}-1) \times 100\%$ (*Nguyen et al., 2017*).

If the 95% confidence interval exceeds 0, it indicates that the application of soil amendments has a significant positive effect on soil pH. Conversely, if the 95% confidence interval is below 0, it suggests that the soil amendments application has a negative effect on soil pH. On the other hand, when the 95% confidence interval spans 0, it means that the soil amendments application has no significant effect on soil pH. The data processing described above was carried out using the metafor package in R software. The significant difference among covariate levels was determined through meta-regression analysis using random effects, maximum likelihood estimator, and mixed-effects models (*Ebensperger, Rivera & Hayes, 2012*; *Ferreira et al., 2016*). Total heterogeneity for a covariate was calculated using mixed-effects models as follows (*Ferreira et al., 2016*):

$$Q_t = Q_m + Q_e \tag{9}$$

where $Q_t$ is the overall degree of heterogeneity estimated by the mixed effects model, with $Q_m$ denoting the total heterogeneity between the levels of covariates explained by the model, and $Q_e$ representing the residuals not explained by the model. If 95% of the CI of the covariates do not overlap, it indicates that the levels of the covariates are significantly different from each other (*Ferreira et al., 2016*). Qm represents the heterogeneity caused by the explanatory variable, and a larger value, indicates a greater influence of the explanatory variable on the effect value. In this study, $I^2 = 99.52\%$, indicating very diverse and highly heterogeneous effect values across different study cases. Furthermore, our data set on amendments' effect on soil pH value showed strong heterogeneity with $Q_t = 39,055.5480$, $p < 0.001$, so it is necessary to introduce explanatory variables to explain the heterogeneity.

## Publication bias test

Meta-analysis is a quantitative assessment of the average effect value, using data obtained from published literature, which may be affected by publication bias. Rosenthal's fail-safe number (Nfs) is used as a tool to assess publication bias in the entire database. If Nfs > 5n + 10 (n represents the sample volume), it indicates the absence of publication bias (*Macaskill, Walter & Irwig, 2001*). In this study, Nfs = 40,896,65, significantly exceeding the threshold of 5n + 10. Therefore, it can be concluded that this study is not affected by publication bias, confirming the reliability of our results.

# RESULTS

## Overall distribution of response ratio to soil pH change in tea plantations

As illustrated in Fig. 1, SPSS26.0 software was utilized to analyze the normal distribution of soil pH change data of the tea plantations treated with soil amendments. The histogram depicts the frequency distribution of the response ratio of soil pH change in the tea plantations, and the fitting curve aligns with a normal distribution ($p < 0.001$), indicating that the data analyzed in the study are consistent and suitable for meta-analysis.

## Effects of different types of amendments on soil pH in tea plantations

Figure 2 illustrates the significant impact of various types of soil amendments on the pH of tea plantations, showing varied levels of increase ($p < 0.0001$). Lime exhibited the highest increase at 18%, followed by animal waste and biochar at 10% and 8%, respectively. In comparison, rapeseed cake and crop straw had a less pronounced effect on the soil pH of tea plantations, with increases of 2% and 5%, respectively.

## Effect of lime on soil pH in tea plantations

Figure 3A illustrates a significant increase in soil pH levels resulting from the application of lime to soils with varying initial pH levels ($p < 0.0001$). When lime was applied to extremely acidic soil (pH ≤ 4.5), the increase in soil pH was 19%, which was higher than that observed in strongly acidic soil (4.5 < pH ≤ 5.5). However, the difference between the two rates was not statistically significant ($p > 0.05$).

Figure 3B shows that lime application has a significant effect on increasing soil pH in tea plantations ($p < 0.0001$); however, the magnitude of this increase varies based on the amount of lime applied. Overall, the rise in soil pH was directly proportional to the amount of lime applied. When the lime application exceeded 4.5 t/ha, the soil pH increase was most pronounced, reaching 27%, a significantly higher value than those observed at M ≤ 1.5 (10%) and 1.5 < M ≤ 4.5 (14%) levels ($p < 0.0001$). Nonetheless, no significant difference was found between M ≤ 1.5 and 1.5 < M ≤ 4.5 ($p > 0.05$). Figure 3C shows the impact of the experiment type on lime treatment to increase soil pH. In the incubation experiment, the increase of soil pH by applying lime was 25%, a significant difference compared to the field experiment ($p < 0.01$).

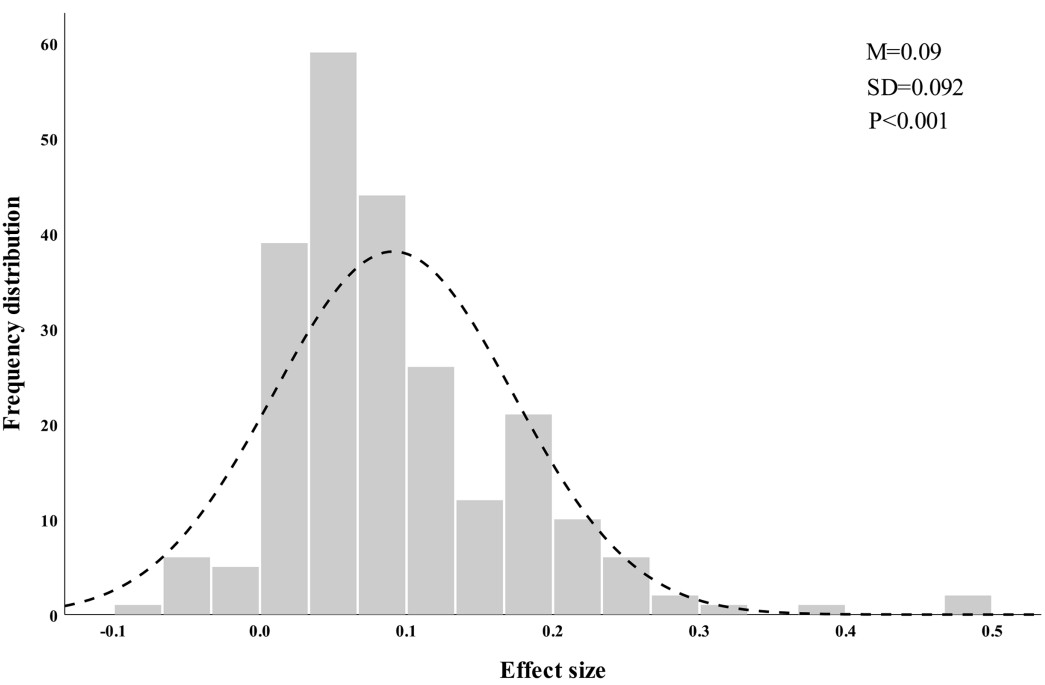

**Figure 1** **Response ratio of tea plantations soil pH to amendments.** M and SD represent the mean and standard deviation of tea plantations soil pH change response ratio, respectively.

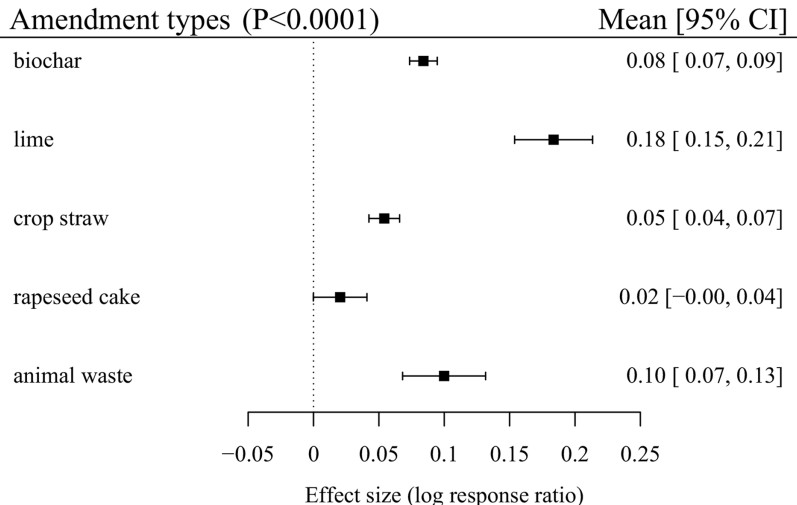

**Figure 2** **Log response ratio of different types of amendments to soil pH in tea plantations.**

## Effect of application crop straw and rapeseed cake on soil pH

Figure 4A illustrates that the application of crop straw can lead to an increase in soil pH in tea plantations. There appears to be a correlation between the quantity of crop straw applied and the rise in soil pH; however, no significant variances were observed among the different application amounts ($p > 0.05$). Applying rapeseed cake at a rate below 10t/ha in

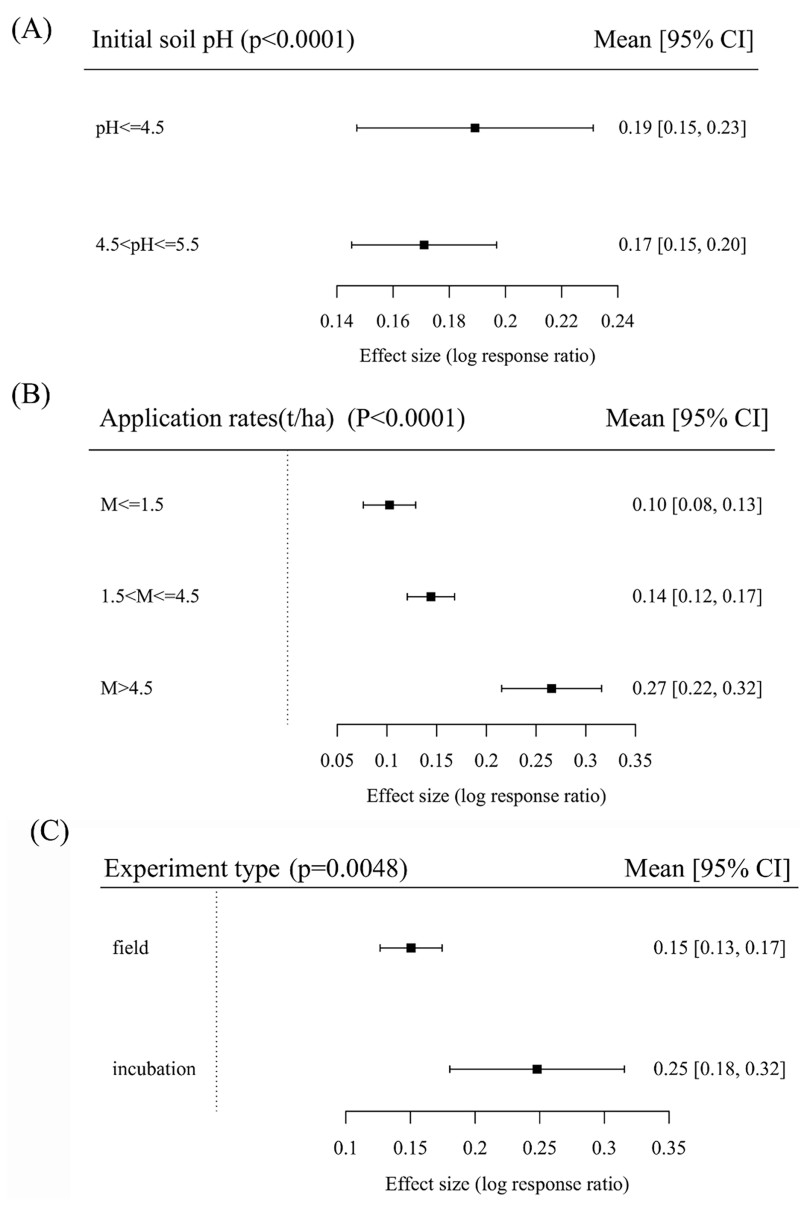

**Figure 3** Log response ratio of initial soil pH (A), lime application rates (B) and experiment type (C) to soil pH in tea plantations.

tea plantations soil can also elevate soil pH, although the increase is less compared to other amendments. Conversely, when the application amount exceeds 10t/ha, it will have an adverse effect on soil pH (Fig. 4B). Figure 4C shows that the application of rapeseed cake in pot experiments resulted in a slightly better increase than was achieved in the field experiments, but the difference was not significant ($p > 0.05$).

## Effect of animal waste on soil pH

Figure 5A demonstrates that the application of varying amounts of animal waste to tea plantations can substantially raise soil pH, with no significant difference in the improvement effect observed between different application rates ($p > 0.05$). Additionally,

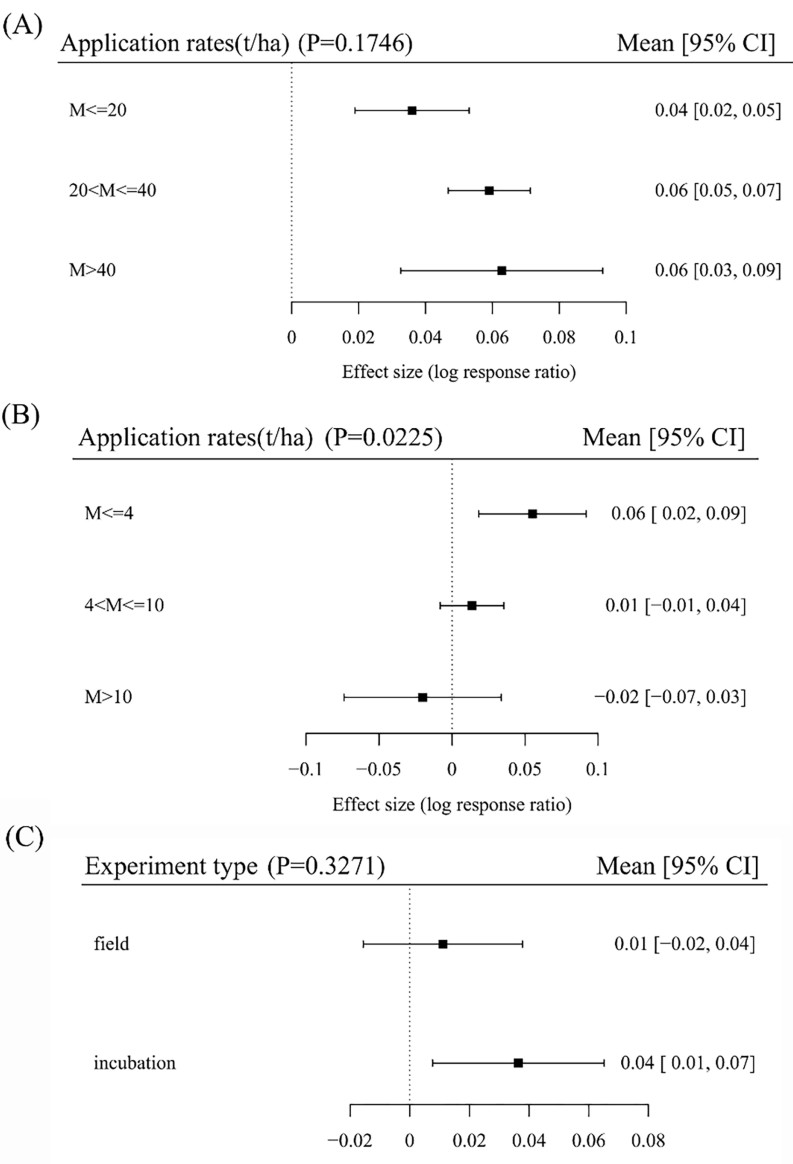

**Figure 4 Log response ratio of crop straw application rates (A), rapeseed cake application rates (B) and experiment type (C) to soil pH in tea plantations.**

applying animal waste with different pH levels in the soil could increase soil pH, and the improvement effect was enhanced with increasing animal waste pH, although the difference was not statistically significant ($p > 0.05$). Furthermore, under different initial soil pH conditions ranged from 5.5 to 6.5, applying animal waste had the most pronounced improvement effect on soil pH, reaching 21%, which was significantly higher compared to initial soil pH levels of 4.5–5.5 and less than 4.5 ($p < 0.05$) (Fig. 5C). In incubation and field experiment, application of animal waste significantly increased soil pH (9% and 10%), but the difference was not significant ($p > 0.05$) (Fig. 5D).

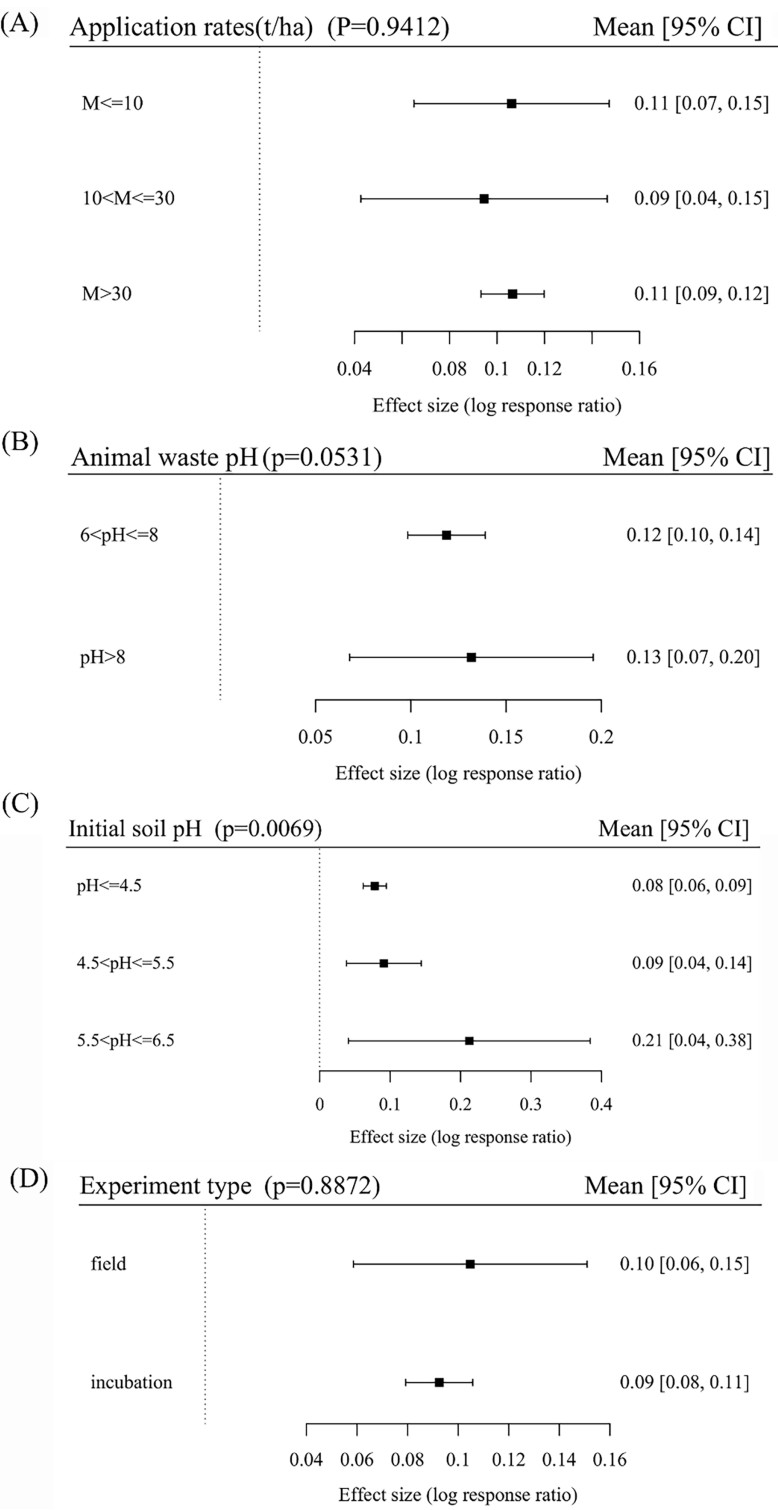

**Figure 5 Log response ratio of animal waste application rates (A), pH (B), initial soil pH (C) and experiment type (D) to soil pH in tea plantations.**

## Effect of biochar on soil pH in tea plantations

### Experimental type

Figure 6A illustrates the impact of different experiment types on the increase of soil pH in tea plantations through biochar treatment ($p < 0.01$). Regardless of experimental type, application of biochar consistently and significantly increased soil pH. In pot conditions, the maximum increase in soil pH due to biochar application was 11%, which was significantly higher than the increases observed in field ($p < 0.0001$) and incubation experiment ($p < 0.05$). Furthermore, the increase in soil pH in field and incubation experiments were 6% and 8%, respectively, with no significant difference ($p > 0.05$).

### Biochar pyrolysis temperature

Figure 6B shows the application of biochar has the most significant effect on increasing the pH of tea plantation soil especially when the pyrolysis temperature ranges from medium to high (501–600 °C), resulting in a 9% increase in soil pH. This effect is followed by a medium temperature range (401–500 °C) and a low temperature range (≤400 °C) with 8% increase. There is no substantial difference in the pH improvement effect on tea plantation soil when biochar is pyrolyzed at medium (401–500 °C), low (≤400 °C), and high temperatures (>600 °C) ($p > 0.05$). Additionally, when the pyrolysis temperature of biochar exceeds 600 °C, the increase in soil pH through biochar application is minimal, at only 6%. Therefore, it can be concluded that the most significant enhancement in soil pH due to biochar application occurs when the pyrolysis temperature ranges from 501–600 °C, followed by the temperature range of 401–500 °C and temperatures below 400 °C.

### Initial soil pH

The application of biochar with a pH range of 5.5 to 6.5 resulted in a significant 11% increase in soil pH, which a larger enhancement compared to that observed in strongly acidic soil (4.5 < pH ≤ 5.5) with an increase of 7% and extremely acidic soil (pH ≤ 4.5) with an increase of 9%. No significant difference was found between the effects on strongly acidic soil and extremely acidic soil ($p > 0.05$). Therefore, the use of biochar in acidic tea plantations soil (5.5 < pH ≤ 6.5) demonstrates a relatively positive impact on increasing soil pH (Fig. 6C).

### Biochar pH

Figure 6D illustrates the impact of applying biochar with different pH levels on soil-pH in tea plantations ($p < 0.001$). As the pH of the biochar increases, the enhancement effect on soil pH gradually intensifies. Notably, biochar with a pH exceeding 10 exhibit the most substantial increase in soil pH, with an enhancement rate of 13%, surpassing those with pH ≤ 8 (6%) ($p < 0.001$) and 8 < pH ≤ 10 (8%) ($p < 0.001$) by 2.17 times and 1.63 times significantly. However, no significant difference was observed in the impact of biochar with pH ≤ 8 and 8 < pH <= 10 on increasing soil pH ($p > 0.05$). The impact of biochar with lower pH levels on increasing soil pH was weaker compared to other pH levels, while biochar with pH levels exceeding 10 exhibited the most significant effect on increasing soil pH. These results indicate that biochar with high pH values has a superior effect on increasing soil pH in tea plantations.

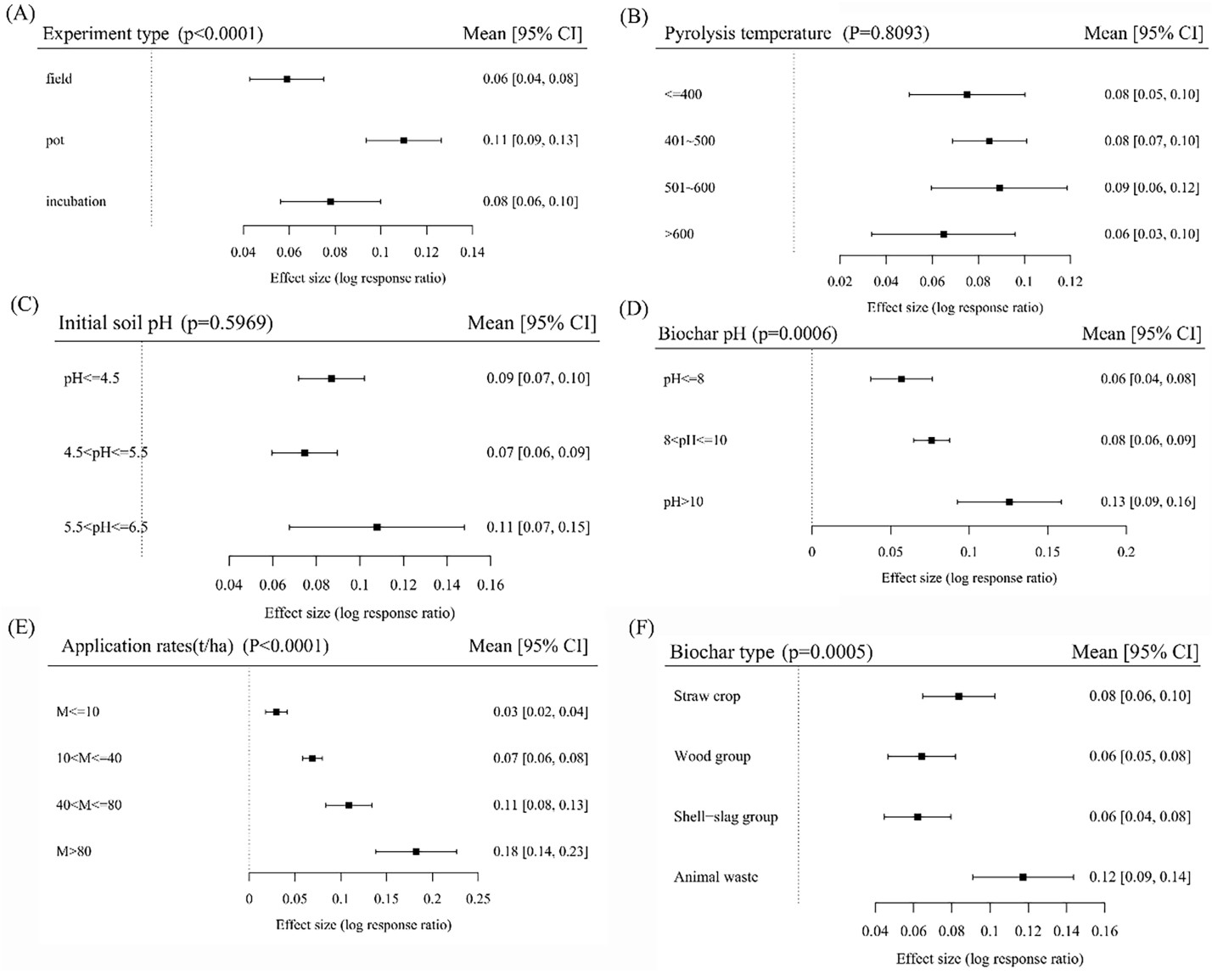

**Figure 6** Log response ratio of experiment type (A), biochar pyrolysis temperature (B), initial soil pH (C), biochar pH (D), biochar application rates (E) and biochar type (F) to soil pH in tea plantations. 

### Biochar dosage

Figure 6E illustrates that varying amounts of biochar application in tea plantations soil result in a significant increase in soil pH levels ($p < 0.0001$). As the amount of biochar amendment increases, the soil pH gradually rises. Specifically, a high dosage of biochar (M > 80 t/ha) resulted in an 18% increase in soil pH improvement, which is six times greater than the increase observed with low dosages (M ≤ 10 t/ha) at 3% ($p < 0.0001$). For biochar application amounts ranging between 10–40 t/ha and 40–80 t/ha, the corresponding soil pH increase is measured at 7% and 11%, respectively ($p < 0.0001$). Therefore, it can be concluded that the application of biochar effectively enhances soil pH, with higher application amounts leading to more pronounced improvements in soil pH levels.

### Raw materials of derived biochar

Biochar derived from various raw materials has been shown to increase the pH of tea plantations soil ($p < 0.001$). Notable, biochar produced from animal waste, such as pig and sheep manure, demonstrates the most significant impact by increasing the soil pH by up to 12%, surpassing other types of biochar ($p < 0.01$). Additionally, crop straw biochar (including wheat straw, corn straw, rice straw, *etc.*) exhibited a notable pH increasing effect, elevating soil pH by 8%, followed by wood biochar (wood, bamboo, tea branches) and shell residue biochar (peanut shell, bamboo residue, coconut shell), which increased the soil pH by 6% ($p > 0.05$).

## Correlation analysis of soil pH and its influencing factors in tea plantations

Within the biochar subgroup analysis, the regression analysis results (Fig. 7) indicated a positive correlation between the soil pH changes and both initial soil pH and biochar pH ($p < 0.01$) in tea plantations. The acidic soil ($5.5 < pH \leq 6.5$) and the application of biochar with higher pH exhibited the most significant effects on soil pH improvement in tea plantations. There were no significant variations in soil pH changes with increasing pyrolysis temperatures of biochar ($p > 0.05$). At 500 °C, the effect on soil pH was unstable resulting in a wide range of changes. Additionally, a positive correlation was found between soil pH changes and biochar application rates ($p < 0.01$). For other types of amendments (Fig. 8), a significant positive correlation was observed between changes in soil pH and the application of lime ($p < 0.01$). However, no substantial correlation was found the application of animal waste, crop straw, and rapeseed cake ($p > 0.05$). Notably, as the amount of rapeseed cake applied increases, the change in soil pH gradually decreases. Overall, it was found that both the pH and type of biochar, as well as the amount of biochar and lime applied, have a greater influence soil pH change in tea plantations. Utilizing animal waste for pyrolysis, alkaline biochar at temperatures ranging from 501–600 °C, and applying lime are more effective on improving soil pH in acidic soils.

## DISCUSSION

Soil acidification in tea plantations has become a global problem, and soil pH has become an important limiting factor for tea tree growth and tea quality. Therefore, optimizing soil pH and controlling soil acidification are crucial for the sustainable development of tea plantations. The results of this study indicate that the utilization of various soil amendments influences soil pH modification, albeit to varying extents. Among them, lime application had the most significant enhancement in reducing soil acidity within tea plantations, resulting in a 18% increase. The primary mechanism behind this phenomenon is attributed to the neutralization of the soil solution by the application of alkaline lime. This process involves the neutralization of excessive acidic ions, including protons and other acidic mineral cations such as $Al^{3+}$, while simultaneously introducing alkaline cations such as $Ca^{2+}$ and $Mg^{2+}$ into the root zone. Subsequently, a substantial influx of calcium and magnesium ions into the soil enhances the cation exchange capacity, thereby decreasing the concentration of exchangeable aluminum in the soil (*Li et al., 2019*).

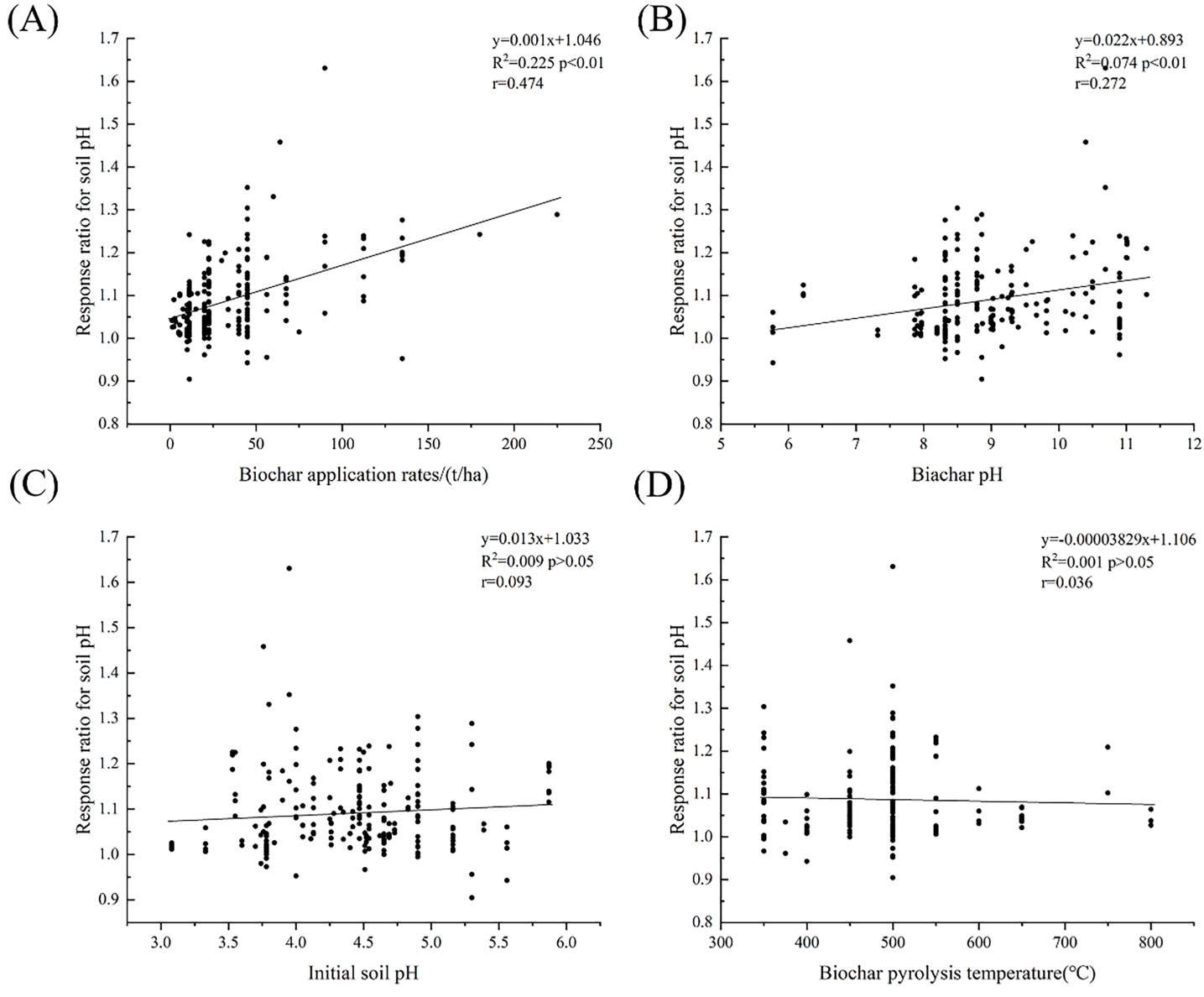

**Figure 7 Relationship between response ratio for soil pH and biochar application rates (A), biochar pH (B), initial soil pH (C), biochar pyrolysis temperature (D).**

In addition, lime application in tea plantations can improve soil physicochemical properties, including soil texture nutrient effectiveness, and so on. Within a certain range, the improvement of the soil is more obvious as the lime dosage increases and the application time is prolonged. Specifically, the presence of calcium in lime materials promotes the formation of soil aggregates, thereby enhancing soil structure (*Bolan, Adriano & Curtin, 2003*). Nonetheless, excessive application of lime has been associated with diminishing effects on soil pH (*Li et al., 2021*). Prolonged and excessive lime use can disrupt the balance of potassium, calcium, and magnesium ions in the soil, damage the soil structure, cause soil compaction, and reduce crop yields (*Haling et al., 2009*).

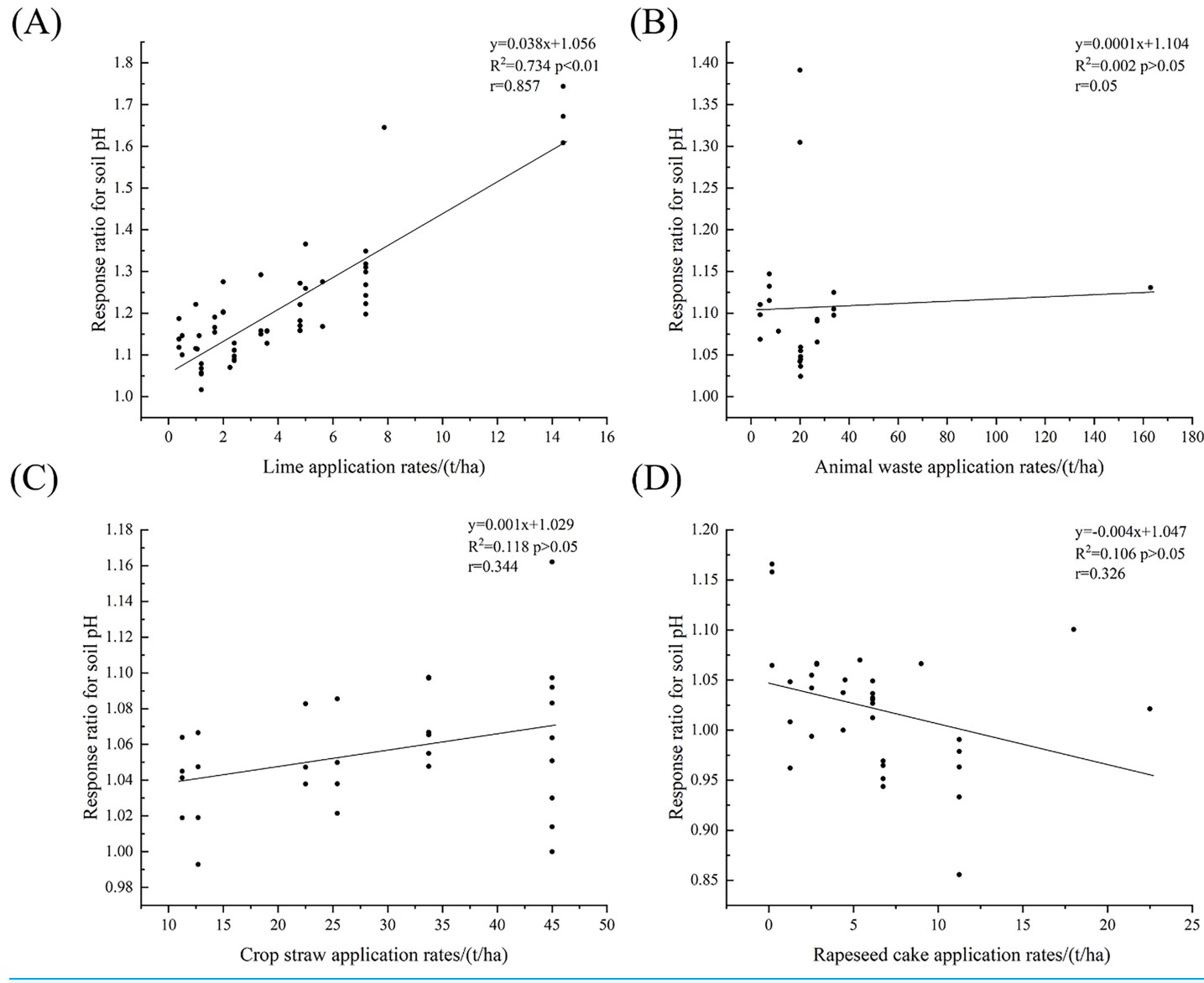

**Figure 8 Relationship between response ratio for soil pH and lime (A), animal waste (B), crop straw (C), rapeseed cake application rates (D).**

As the pH of the modified materials in this study increases, there is a corresponding rise in the pH levels of the tea plantation soil. This phenomenon can be attributed to the higher alkaline content present in the modifiers with elevated pH levels. It allows them to effectively neutralize hydrogen ions within the soil of the tea plantations (*Dai et al., 2014*; *Yuan & Xu, 2010*). Furthermore, the biochar with higher pH in the modifiers contains more ash, which includes potassium, calcium, magnesium and other elements that can improve soil base saturation by absorbing and decreasing soil hydrogen ion and exchangeable aluminum content, thereby reducing soil acidity (*Van et al., 2009*). In this study, the application of soil amendments increased the pH of the tea plantations soil to

varying degrees. This improvement effect amplified with increasing application amount, consistent with previous studies (*Han et al., 2007*; *Liu et al., 2023a*).

The initial pH of biochar plays a critical role in influencing soil pH alterations (*Tan et al., 2019*). This phenomenon is attributed to the base ions present in biochar ash undergoing cation exchange reactions with hydrogen ions and exchangeable aluminum in the soil. Additionally, the substantial presence of negatively charged hydroxyl and carboxyl groups in biochar facilitates their neutralization with $H^+$ in the soil, thereby contributing to the increase in pH in tea plantations soil (*Chintala et al., 2014*). The research results indicate that biochar derived from various raw materials has the potential to enhance the pH of soil in tea plantations. The increase observed in biochar derived from animal waste materials (12%) surpasses that of biochar from other sources, consistent with the results reported by *Geng et al. (2022)*. This is attributed to the higher pH and ash content in biochar derived from animal waste (*Geng et al., 2022*; *He, Luo & Zhu, 2024*).

The properties and characteristics of biochar are dependent on the pyrolysis temperature at which it is produced (*Yang & Lu, 2021*). *Jiang et al. (2021)* investigated the impact of different pyrolysis characteristics including surface area and surface morphology. Their findings revealed that the average pore radius was approximately 0.91 nm greater at higher pyrolysis temperature, while the BET surface area was 1.74 $m^2$/g lower at elevated pyrolysis temperature. Furthermore, they found that biochar produced at lower temperatures exhibited a smoother surface and fewer pores. As the pyrolysis temperature increased, the surface roughness of biochar gradually heightened, marking the pore structure more pronounced. This study also observed that as pyrolysis temperature increased, there was an intensified enhancement effect on the pH of tea plantations soil with an increase in biochar pyrolysis temperature. The biochar pyrolysis temperature of 501–600 °C exhibited the most substantial enhancement in the pH of tea plantations soil. However, when temperatures exceed 600 °C, the improvement effect notably decreases. This is consistent with the research results of *Yang et al. (2021)* and *Geng et al. (2022)*, which suggest that biochar pyrolysis temperature at 501–600 °C have a greater influence on soil pH compared to other pyrolysis temperatures. In general, as the pyrolysis temperature increases, the pH of most biochar also increases, leading to continuous enrichment of inorganic elements. This is attributed toa corresponding increase in the degree of dehydration and decomposition of organic acids in organic matter, as carbonization temperature rises, resulting an increase in basic groups (*Geng et al., 2022*). Furthermore, as the temperature of carbonization increases, the levels of K, Na, Ca, Mg, and other mineral elements in the biochar oxides and carbonates also increase (*Das, Ghosh & Avasthe, 2020*). However, when the pyrolysis temperature exceeds 600 °C, the hydroxyl content on the surface of biochar decreases, and the phenolic hydroxyl and carboxyl groups gradually weaken or even disappear as the temperature rises further, weaken the improvement effect on soil pH (*Geng et al., 2022*; *He, Luo & Zhu, 2024*). *Jiang et al. (2021)* demonstrated that the soil pH enhancement effect of biochar pyrolysis at 350 °C surpassed that at 500 °C when the identical amount of biochar was applied, potentially attributed to variations in crop types and biochar raw materials.

The findings of this study suggest that biochar applied to acidic soils ($5.5 < \text{pH} \leq 6.5$) are effective in raising the pH value, demonstrating an increase of 11%. Acidic soils, characterized by lower pH values, exhibit a significant presence of acidic ions such as hydrogen ions ($H^+$). Soil amendments such as lime, biochar, have the capability to neutralize acidity through chemical reactions with these acidic ions, leading to an elevation of the soil's pH value. Furthermore, certain amendments have the capacity to release bicarbonate ions ($HCO_3^-$) and hydroxides ($OH^-$), further enhancing the soil's pH value (*Wang et al., 2017*). However, when biochar is applied to extremely acidic and highly acidic soils, only a slight increase in pH is observed. This phenomenon could be attributed to the increased in soil acidity and lower pH levels, leading to a significant rise in $H^+$ concentration and elevated levels of active acids in the soil. As a result, the introduction of biochar strengthens, the soil's resistance strengthens, causing a slower decrease in soil $H^+$ concentration over time (*Lieb, Darrouzet-Nardi & Bowman, 2011*). Furthermore, studies have shown that the application of biochar in strongly acidic soils with pH levels between 4.5 and 5.5 has a notable impact on soil pH. This can be linked to the natural preference of tea trees for acidic soil, as well as the varying degrees of tolerance to soil acidity and sensitivity to pH levels among different crop species. Moreover, these discrepancies may arise from differences in crop and material types, as well as regional climate variations.

## CONCLUSIONS

In summary, the addition of amendments with higher pH levels ($5.5 < \text{pH} \leq 6.5$) of acidic soil in tea plantation yields a more significant improvement in soil quality. Furthermore, there exists a significant positive correlation between the amount of various amendments and the increase in soil pH ($p < 0.05$). Among these amendments, lime application stands out as the most effective method for modifying soil acidity, but the scientific application of lime is crucial for improving soil acidification. Additionally, within the realm of biochar, utilizing animal manure as a raw material and producing biochar with a pyrolysis temperature ranging from 501 °C to 600 °C demonstrates superior efficacy in improving soil quality at tea plantations. Our research provides a scientific basis for enhancing the soil quality of tea plantations, improving tea quality, and ensuring the long-term health and sustainability of tea plantations.

## ACKNOWLEDGEMENTS

We thank all the researchers whose data were used in this meta-analysis. Thanks to all contributors in this study.

### Funding

This work was financed by the National Natural Science Foundation of China (32371543), and the Key Research and Development Project of Science and Technology Department of Zhejiang Province (2022C02022, 2023C02020). The funders had no role in study design, data collection and analysis, decision to publish, or preparation of the manuscript.

## Grant Disclosures

The following grant information was disclosed by the authors:
National Natural Science Foundation of China: 32371543.
Key Research and Development Project of Science and Technology Department of Zhejiang Province: 2022C02022, 2023C02020.

## Competing Interests

The authors declare that they have no competing interests.

## Author Contributions

- Zhenyu Yang conceived and designed the experiments, performed the experiments, analyzed the data, prepared figures and/or tables, and approved the final draft.
- Bo Yao performed the experiments, prepared figures and/or tables, and approved the final draft.
- Ronghui Li analyzed the data, prepared figures and/or tables, and approved the final draft.
- Wenyan Yang conceived and designed the experiments, analyzed the data, prepared figures and/or tables, and approved the final draft.
- Dubin Dong analyzed the data, authored or reviewed drafts of the article, and approved the final draft.
- Zhengqian Ye conceived and designed the experiments, authored or reviewed drafts of the article, and approved the final draft.
- Yuchun Wang conceived and designed the experiments, prepared figures and/or tables, authored or reviewed drafts of the article, and approved the final draft.
- Jiawei Ma conceived and designed the experiments, analyzed the data, prepared figures and/or tables, authored or reviewed drafts of the article, and approved the final draft.

## Data Availability

This is a systematic review/meta-analysis.

## Supplemental Information

Supplemental information for this article can be found online at http://dx.doi.org/10.7717/peerj.17653#supplemental-information.

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
