# Peer review of "Systematic review assessing the effects of amendments on acidic soils pH in tea plantations"

_PeerJ, doi:10.7717/peerj.17653_

## Round 0.1 · original submission · Major Revisions

Reviewers have now completed evaluation of your manuscript. I concurr with their suggestion and invite you to revise your manuscript before it could be accepted for publication.

**Language Note:** The review process has identified that the English language must be improved. PeerJ can provide language editing services - please contact us at [email protected] for pricing (be sure to provide your manuscript number and title). Alternatively, you should make your own arrangements to improve the language quality and provide details in your response letter. – PeerJ Staff

Reviewer 1 ·

Basic reporting

A total of 58 literatures on the effect of improvement materials on the pH of tea plantations soil were obtained through literature search. the effect of applying improvement materials on increasing the pH of tea plantations soil and its influencing factors were explored. There are some issues with the article, and it is recommended to make major revisions.
1. The title does not match the research content. The title is "Effects of different improvement materials on acidified soil improvement in tea plantations ", but the content of the article only involves increasing pH. Acidic soils not only suffer from low pH but also from issues such as toxic heavy metal pollution and active aluminum leaching.
2. The data analysis in the article was not thorough. As for the relevant analysis in the chapter, I don't know which factor has a greater impact, it is just a suggestion for readers.
In the relevant analysis of 3.7, if it is possible to tell readers which properties of the modifier have a greater or lesser impact on pH, the effect is better.
3. The logic in the text was unclear. 3.5 includes different modifiers, why is biochar listed separately in 3.6?
4. There are many errors in the article. For example, in Chapter 3.7, Fig.9 is referred to as Fig.8, and the title and content labeling of Fig.9 are incorrect. Suggest checking and modifying the entire text.

Experimental design

no comment

Validity of the findings

no comment

·

Basic reporting

The manuscript titled "Effects of Different Amendments on Acidified Soil Improvement in Tea Plantations: A Meta-Analysis" by Yang et al. presents a commendable effort to synthesize existing literature on the liming potential of various soil amendments in tea plantations, aligning well with the journal's scope. However, the manuscript requires significant revisions to enhance its quality.

Major Comments:

1. Writing:
The manuscript requires thorough language editing. For instance, the word "improvement" is excessively used 11 times in the abstract and twice in the title, indicating a lack of proficiency in scientific writing.

Experimental design

Data:
The author's reliance solely on literature from China is problematic. Given the global issue of soil acidification in tea plantations, all available data from diverse sources, including unpublished studies, should be included to prevent publication bias.

Categorization:
Several limitations exist in the categorization approach adopted for this meta-analysis:

The type of study was overlooked. Soil incubation, pot, and field studies have varying implications and should be categorized accordingly.
Grouping all soil amendments based on their initial pH without considering their nature lacks scientific rationale. Sub-categories such as experiment type, application rates, pH, and raw materials should be applied separately to each soil amendment group. Soil buffering capacity should also be considered, as it can influence liming effects.

Validity of the findings

No comment

Additional comments

Minor Comments:

Modify the title to avoid repetition, e.g., "Systematic Review Assessing the Effects of Amendments on Acidic Soils in Tea Plantations."

L19 Include an introductory sentence in the abstract addressing soil acidification in tea plantations.

L19 Replace "literatures" with "original research articles" in L19.

L20-22 Use "soil amendments" instead of "improvement materials" and "non-improvement material"

Remove "our" from L38.

Use "less than" instead of "lower" in L40.

Clarify the relationship between acidification and soil structure in L40-45, and provide additional citations.

L50 Introduce different approaches for soil liming under tea plantations before elaborating on each approach .

L54-58 Consider broader literature coverage for biochar effects on soil physico-chemical characteristics

Acknowledge all contributors in the acknowledgment section, not within the manuscript text (L89-90).

L191 what is full text assessment?

L123 Use t/ha instead of Kg/hm2 .

Move L268-286 to the introduction section.

L296 Discuss the potential influence of liming on soil texture.

L318 Explain why biochars derived from animal waste exhibited higher liming effects .

L323-328 Does the mentioned biochar characteristics i.e. BET, surface morphology, pore radius play role in its soil liming ability.

L334 Address why high-temperature biochar did not show liming effects despite enhanced characteristics

Reviewer 3 ·

Basic reporting

This is an interesting topic. but english writing is still needed further revision,and attention must be paid to English grammar. References for meta-analysis still have some limitations, such as some studies on the improvement of soil by leguminous green manure. Most of the figures were not statistically analyzed.

Experimental design

Experimental design must highlight the randomness and universality of sampling. Therefore, references for meta-analysis need to be improved.

Validity of the findings

The validity of the research results still needs to be tested by relevant experiments.

---

## Round 0.2 · Minor Revisions

Please revise your manuscript in the light of suggestions made by reviewers.

**Language Note:** The review process has identified that the English language must be improved. PeerJ can provide language editing services - please contact us at [email protected] for pricing (be sure to provide your manuscript number and title). Alternatively, you should make your own arrangements to improve the language quality and provide details in your response letter. – PeerJ Staff

·

Basic reporting

Revised version have thoroughly improved however manuscript can be further improved.

1. A thorough English editing, English grammar used in the text isnt up to the mark.

2. Scientific editing _ some of the crucial points either need correction or elaboration
For instance, the optimal pH required for tea plantations ranged between 4-5-5.5, so discussion should revolve around optimizing the soil pH for tea plantation or should it be related with liming effect? this point is necessary and should be discussed with some expert.
In response to my previous comment on liming effect on soil texture, author wrote `...lime in tea plantation can also enhance the soil physical structure such as soil texture, structure and nutrient availability`. categorizing the soil texture as soil physical structure points lack of command on basic soil science.
I highly recommend for an internal review of the manuscript from a soil science expert.

3. Quality of figures can be improved. Figure 7 and 8, Y axis should be response ratio instead of change in soil pH

Experimental design

some unit unification appraoches require reference i.e. soil pH via CaCl2 to H2O, rates of soil amendments in pots and incubation.

a category based on soil incubation, pot exp and field exp would be helpful as presence of plants could either reduce or trigger the liming effects.

Validity of the findings

Ok

Additional comments

good effort

Reviewer 3 ·

Basic reporting

The article should include sufficient introduction and background to demonstrate how the work fits into the broader field of knowledge. I always think that the literature on the improvement of tea garden soil pH was not comprehensive enough. Can we find some related research on the improvement of tea garden soil by leguminous green manure ?

Experimental design

Only Web of Science ( https://www.webofscience.com/wos), and the China National Knowledge Infrastructure ( https://www.cnki.net/) to investigate ) was used. I think there should be other common databases that can be used to investigate, so please adding others databases to ensure the integrity of the investigation.

Validity of the findings

No comment

---

## Round 0.3 · accepted · Accept

Revision is done up to satisfactory level and I recommend this paper for acceptance and publication.